# Ab Initio Molecular Dynamics Simulations of the Interaction between Organic Phosphates and Goethite

**DOI:** 10.3390/molecules26010160

**Published:** 2020-12-31

**Authors:** Prasanth B. Ganta, Oliver Kühn, Ashour A. Ahmed

**Affiliations:** 1Institute of Physics, University of Rostock, Albert-Einstein-Str. 23-24, D-18059 Rostock, Germany; prasanth.ganta@uni-rostock.de (P.B.G.); oliver.kuehn@uni-rostock.de (O.K.); 2Department of Life, Light, and Matter (LLM), University of Rostock, Albert-Einstein-Str. 25, D-18059 Rostock, Germany

**Keywords:** P–inefficiency, goethite, glycerolphosphate, inositol hexaphosphate, MD simulations, QMMM, binding energies, infrared spectra

## Abstract

Today’s fertilizers rely heavily on mining phosphorus (P) rocks. These rocks are known to become exhausted in near future, and therefore effective P use is crucial to avoid food shortage. A substantial amount of P from fertilizers gets adsorbed onto soil minerals to become unavailable to plants. Understanding P interaction with these minerals would help efforts that improve P efficiency. To this end, we performed a molecular level analysis of the interaction of common organic P compounds (glycerolphosphate (GP) and inositol hexaphosphate (IHP)) with the abundant soil mineral (goethite) in presence of water. Molecular dynamics simulations are performed for goethite–IHP/GP–water complexes using the multiscale quantum mechanics/molecular mechanics method. Results show that GP forms monodentate (**M**) and bidentate mononuclear (**B**) motifs with **B** being more stable than **M**. IHP interacts through multiple phosphate groups with the **3M** motif being most stable. The order of goethite–IHP/GP interaction energies is GP **M** < GP **B** < IHP **M** < IHP **3M**. Water is important in these interactions as multiple proton transfers occur and hydrogen bonds are formed between goethite–IHP/GP complexes and water. We also present theoretically calculated infrared spectra which match reasonably well with frequencies reported in literature.

## 1. Introduction

Phosphorus (P) scarcity is becoming one of the major global environmental challenges, which needs attention on par with climate change because of the foreseen P peak scenario [1,2,3]. The P in today’s fertilizer is mostly from mined P rocks, and given the current mining rate they are predicted to be exhausted within next 50–100 years [4,5]. Complicating the situation further, the P rock reserves are available only in few countries and a P peak scenario could affect regions like UK, Western Europe, and India who obtain these resources mainly through imports [2,6,7]. As P rock reserves dwindle with time, the food security of P-importing countries is in question, which, in the long-term, might flip oil based economies to P based economies. The oil crisis in the 1970s emphasized the need for renewable energy sources, but unfortunately for the peak P crisis there is no alternative but to develop ways to increase, reuse, and secure the domestic P production [1,6,7,8].

The P input to soil (P fertilizers or from nature) is not fully available to plants as most of it is bound to soil organic matter [9,10,11] or soil minerals [12,13,14,15,16]. To overcome this, fertilizers are often applied in the agriculture industry to maintain and boost the agricultural production. As a side effect, P bound to minerals runs off along water paths during heavy rains and in the long-term causes eutrophication of waterways [17]. Recent studies show that the soil minerals in rain water retain about 50% of P in the soil solution [18,19]. In addition, heavy rains further decrease P efficiency as P-bound soil minerals, i.e., P colloid complexes, disperse with rain water and accumulate in specific regions unavailable to plants [20]. Methods that support effective extraction of P from these colloids would increase domestic P sources which in turn would improve the food security of global population.

Orthophosphate (OP) is one of the most abundant inorganic phosphate, which exists mainly in the form of phosphate ions ($H_{2}{\mathrm{PO}}_{4}^{1-},{\mathrm{HPO}}_{4}^{2-},{\mathrm{PO}}_{4}^{3-}$) [21,22]. Regarding organic phosphates, inositol hexaphosphate (IHP) [23,24,25] and glycerolphosphate (GP) [26,27,28] are some of the abundant and most common organic phosphates in soil. Often they are present in soil as oxyanions with deprotonated phosphate groups [29,30,31]. Therefore, the negatively charged phosphates bind to positively charged soil mineral surfaces to form P colloids. The most common P-fixing minerals are Fe and Al(oxyhydr)oxides [32,33,34] and Ca–oxides [33]. Goethite (α-FeOOH) is one of the most reactive and abundant mineral that interacts with phosphates [35]. Strong interactions have been reported for OP [15,36,37,38,39] as well as for IHP [30,31,40,41] and GP [29], especially at low to medium pH. The interaction of phosphates with goethite or most minerals is through bonded and non-bonded interactions, for instance, covalent bonds, hydrogen bonds (HBs), van der Waals (vdW) interactions, and electrostatic attraction to form surface complexes [33,42,43]. Numerous studies show that P compounds with a single phosphate group form one to two covalent bonds between the phosphate group’s oxygens and surface iron atoms [15,36,44] within inner-sphere complexes. IHP may bind through one to four of its six phosphate groups [31,40,41,45] to form covalent bonds with surface iron atoms. In contrast, Johnson et al. [30] suggested that IHP forms outer-sphere complexes. In addition to the covalent bonds and electrostatic interaction between phosphates and goethite, water also plays an important role for the stability of these complexes. In fact, the P colloids often exist in a solvated state in both arable and forest soils. The study by Ahmed et al. [46] showed the importance of water in surface complexation reactions and highlighted how water maneuvers the phosphate interaction with mineral by forming strong to moderately strong HBs with phosphates [47,48].

A molecular level study of solvated P colloids could provide additional insight into these interactions and improve our understanding of P availability to plants [13]. Molecular simulations are efficient tools to achieve this with proven track record [42,49]. For instance, Kwon and Kubicki [50] resolved controversies in experimental studies related to phosphate surface complexes on iron hydroxides. They correlated different types of phosphate binding motifs with goethite to pH and suggested that binding motifs change with pH. Aquino et al. [51] estimated goethite’s point of zero charge (PZC) to 9.1 which fits well with experimental values of 6.4–9.7 [22,52,53]. Kubicki et al. [44] explored phosphate and goethite binding motifs and their corresponding stability at different surface planes to find dominant motifs that contribute significantly to infrared (IR) spectra. Ahmed et al. [15] showed that an analysis of theoretical spectra for different motifs could provide an estimate of their ratio of abundance at the goethite surface.

The main objective of the current study is to unravel the interactions of IHP/GP at the 010 goethite–water interface. This is achieved by analyzing binding motifs, interaction energies, and IR spectra obtained using periodic boundary quantum mechanics/molecular mechanics (QM/MM) [54] molecular dynamics simulations.

## 2. Modeling and Computations

### 2.1. Goethite Surface

Goethite is an orthorhombic crystal containing ferric (${\mathrm{Fe}}^{3+}$) iron which is coordinated by six oxygen atoms [35]. The bulk oxygens are triply coordinated and they are of two distinct types: (1) bridging oxygen coordinated by three iron atoms plus a HB and (2) hydroxyl oxygens coordinated by three iron atoms and one proton, see Figure 1c. Its unit cell contains 16 atoms, i.e., four FeO(OH) units with lattice constants a= 9.95, b= 3.01, c= 4.62 Å. Some common goethite surface planes are 010, 100, 021, and 110 (as per Pnma space group). The IHP and GP interactions with goethite are studied here by considering the 010 goethite surface which is known for its high stability [55,56], see Figure 1c. The unsaturated and undercoordinated iron and oxygen atoms at the goethite surface generate an overall positive surface charge that attracts IHP/GP and also water to the surface. Compared to bulk, the surface iron atoms are coordinated by only four oxygens while the bridging and hydroxyl oxygens are coordinated by only two irons instead of three. Modeling an undercoordinated surface supports experimental studies that phosphate (IHP [31] and GP [29]) adsorption occurs mostly at pH lower than the goethite PZC, where its surface is unsaturated and has positive surface charge [29,31].

### 2.2. Model Systems

First, a surface slab is modeled by repetition of the goethite unit cell as 2a× 4b× 5*c*. Then, GP (see Figure 1a) is added onto the goethite slab to form covalent bond(s) with a surface iron atom, see Figure 1c. The binding motifs considered for goethite–GP complex are based on experimental studies related to OP [22,57] and organic compounds with monophosphate groups [29,43] interacting with goethite. The study by Li et al. [29] of GP adsorption on goethite suggests that GP predominantly forms monodentate mononuclear motifs (**M**, 1Fe + 1O a covalent bond between Fe atom and phosphate free oxygen, see Figure 1d). However, considering OP’s [58] interaction with goethite, an additional motif called bidentate mononuclear motif (**B**, 1Fe + 2O two covalent bonds between Fe atom and unprotonated and protonated phosphate O atoms, see Figure 1e) is modeled. The bidentate binuclear motif (**BB**, 2Fe + 2O, two covalent bonds between two surface Fe atoms and two oxygens from phosphate, see Appendix A) which is commonly discussed for OP interaction with goethite [22,36,50] is not considered here as the distance between consecutive Fe atoms does not match the range of distances between oxygen atoms in the phosphate group.

Similarly, IHP (see Figure 1b) is added onto the goethite surface to form goethite–IHP complexes. The IHP interaction with goethite is different compared to GP because of the additional phosphate groups. Guan et al. [45] studied IHP adsorption onto aluminum hydroxide and showed that IHP binds to the surface through two phosphate groups. Moreover, Celi et al. [31] and Ognalaga et al. [40] suggested that IHP interacts with goethite through multiple phosphate groups. Unlike for OP and GP, there is no suggestion for a clear dominant motif for IHP interaction with goethite. Therefore, in addition to the **M** and **B** motifs, a tridentate motif **3M** (**3M**, 3Fe + 3O three covalent bonds between three surface Fe atoms and one protonated, two unprotonated phosphate O atoms, see Figure 1f) is considered. The **3M** motif is considered here based on the goethite surface Fe atom’s and IHP O atom’s structural flexibility to form bonds.

The complexes discussed so far are solvated to include the effects of water on their interactions. The goethite–IHP/GP complexes are solvated perpendicular to the studied surface plane up to ≈ 18 Å using VMD [59] at a density of ≈ 1 $g{\mathrm{cm}}^{-3}$, see Figure 1c. To ease the discussion about the interactions, the Fe/Al-bonded oxygens are denoted as $O_{p}$. Note that even though the initial motifs considered here may not include all possible surface configurations they should allow us to draw conclusions from common motifs.

### 2.3. Computational Details

The atoms in the modeled complexes are separated into two regions as required by the QM/MM method [54,60]. The atoms of the reactive region (QM) of goethite–IHP/GP–water complexes include the top goethite layer, phosphate, and water within ≈ 10 Å from goethite surface. The average number of QM atoms in each model is around 350–400 depending on the complex; the total average number of atoms per each model is around 1200–1250. The atoms excluded from the QM region are described at the MM level with classical force fields (FF). Note that simulating the above complexes with 1200+ atoms using pure QM methods is a computational roadblock [42,61], and to overcome this the QM/MM approach is adopted here.

The potential and forces of the QM subsystems are calculated using density functional theory (DFT) as implemented in the Quickstep code [62]. Here, the nucleus and highly localized core electrons of atoms are replaced by Goedecker–Teter–Hutter (GTH) pseudopotentials [63], while the valance electrons are described with the double-ζ valance-polarized MOLOPT (DZVP–MOLOPT–SR–GTH) basis set [64]. For water atoms, the valance electrons are defined with the simpler single-ζ valance (SZV–MOLOPT–SR–GTH) basis set to further reduce computational cost. The exchange correlation interactions and vdW interactions are included with Perdew–Burke–Ernzerhof (PBE) [65] exchange correlation functional and D3 empirical dispersion correction [66], respectively. The MM part is simulated with the FIST module [67], which is an integral part of CP2K [68,69]. The goethite surface is modeled with the CLAYFF FF [70] and the water with the single point charge (SPC) water model [71]. For phosphates IHP and GP, CHARMM FF are obtained from SwissParm, a FF generation tool [72]. Both CLAYFF and CHARMM FFs are compatible with the SPC water model.

The interaction between QM and MM subsections is simulated by Gaussian expansion of the electrostatic potential method (GEEP) [73] which is part of CP2K. This method defines MM charges as smeared out Gaussians and adopts efficient screening techniques to reduce the computational cost in calculating mutual interaction energies between QM and MM subsystems. There is a variety of QM/MM methods available that differ based on the level of theory used to define QM and MM mutual interactions. Here, the electrostatic embedding type of QM/MM method is employed which allows MM atoms to polarize QM atoms and thus to include the effect of surrounding atoms onto the reactive region [60]. For all complexes, the QM box size is 2a× 22× 5*c*, i.e., 19.9 × 22 × 23.1 Å, see Figure 1c, and the remaining region is MM.

Molecular Dynamics (MD) simulations are employed to sample the equilibrium dynamics including reactive events. As the phosphate interaction at goethite–water interface involve proton transfer events and covalent bond changes [46,47,48], using the QM/MM MD technique is mandatory. Specifically, the electron density cut-off for the auxiliary plane wave basis set is chosen to be 500 Ry with SCF convergence threshold of $10^{-4}$ Hartree. The MD simulations are performed for 25 ps with a 0.5 fs time step and the temperature is maintained at 300K with canonical sampling through the velocity rescaling thermostat (CSVR) [74].

Each model has been equilibrated for about 10 ps. Interaction energies and information about geometries have been obtained from a subsequent 15 ps production trajectory. Specifically, the interaction energy between the goethite surface and IHP is calculated for every 100 fs (i.e., 150 snapshots) along the production trajectory by using
(1)$$E_{\mathrm{int}}=E_{\mathrm{goe}-\mathrm{IHP}-\mathrm{complex}}-\left(E_{\mathrm{goe}}+E_{\mathrm{IHP}}\right)\;.$$

The terms $E_{\mathrm{goe}-\mathrm{IHP}-\mathrm{complex}}$, $E_{\mathrm{IHP}}$, and $E_{\mathrm{goe}}$ denote total electronic energy of the goethite–IHP–complex, IHP, and goethite surface, respectively. The basis set superposition error (BSSE) in interaction energies is corrected using the counterpoise scheme [75]. The interaction energies that involve water are divided by the total number of water molecules in the simulation box to obtain per water molecule interaction energies for better comparison.

Infrared (IR) spectra for IHP and GP adsorbed onto goethite are calculated using the TRAVIS [76] software. In TRAVIS the IR spectra are obtained by Voronoi tessellation of the electron density yielding molecular dipole moments from bulk phase MD simulations [77,78]. This was done along a 30 ps equilibrium trajectory. The electron density is calculated for each 4 fs, i.e, every 8th snapshot of production trajectory and the massive (≈ 2 TB) electron density files are compressed using the bqb compression tool [79] compatible with TRAVIS. The compressed bqb files are then provided to TRAVIS software to calculate IR spectra. Here, the IR spectra are calculated for the frequency range of 950 to 1250 ${\mathrm{cm}}^{-1}$, where the characteristic peaks related to phosphate stretching modes are observed.

## 3. Results and Discussion

### 3.1. Goethite–GP–Water Interactions

#### 3.1.1. GP **M** Motif

A stable **M** motif is observed throughout the production trajectory with an average Fe–$O_{p}$ bond length of 2.0 Å and average Fe–P distance of 3.2 Å, see Figure 2b. During early stages of equilibration, GP has formed an average of seven HBs with water, of which one is strong enough for a proton transfer to happen from GP to water, see Figure 3a. By the end of the equilibration, GP is twice deprotonated with one proton transfer to water and another to surface, see Figure 3a–b. A proton transfer is observed from GP to water (see Figure 3c) in the production trajectory; therefore, GP is three times deprotonated and its phosphate group completely deprotonated. In Figure 3b, a proton transfer event is observed from goethite surface oxygen to water. On average 8 HBs are observed between GP and water over the course of the production trajectory, see Appendix A. The HBs are calculated using the VMD [59] plugin with HB donor–acceptor distance selected as 3 Å. Figure 3d shows a stable GP **M** motif at 25 ps.

The average interaction energy between goethite and GP calculated using Equation (Equation 1) is −112 kcal/mol, see Figure 2a. The average GP–water interaction energy per water molecule is −2.3 kcal/mol. The accumulation of water around the goethite surface suggests that water experiences a strong electrostatic pull from goethite surface, see Appendix A. The average goethite–water interaction energy is −6.8 kcal/mol, which is larger than the GP–water interaction energy. The stronger goethite–water interaction is due to multiple Fe–$O_{H_{2}O}$ covalent bonds (see Appendix A), proton transfers, and HBs between goethite active sites and water.

#### 3.1.2. GP **B** Motif

The initial **B** motif was found stable during the production trajectory with a mean average Fe–$O_{p}$ bond length of 2.21 Å and Fe–P distance of 2.6 Å. The mean average Fe–$O_{p}$ bond length observed here is longer than in the **M** motif case, probably due to the repulsion between the bonded oxygens, see Figure 2b. In contrast to the **M** motif, proton transfer events are not observed from GP to the goethite surface. GP formed an average of eight HBs with water during equilibration stage and of these two yielded proton transfers from GP to water, see Figure 3e–f. Therefore, GP is deprotonated twice at the end of equilibration and during the production trajectory it formed an average of eight HBs with water. The snapshots in Figure 3g and h show the GP **B** motif at 20 and 25 ps, respectively.

The goethite–GP per bond interaction energy here is −61 kcal/mol per bond which is less than for the GP **M** motif because of the additional proton transfer observed in the latter case. However, the total interaction energy (−122 kcal/mol) here is higher than for the GP **M** motif, see Figure 2a. The GP–water interaction energy is −1.9 kcal/mol, which is smaller than for **M** motif case, probably due to the additional proton transfer from GP to water in the **M** motif.

#### 3.1.3. Discussion of Goethite–GP–Water Interactions

The study by Li et al. [29] of GP on goethite suggested the formation of only the **M** motif and considered that the **B** motif could not be formed due to steric hindrance of the organic moiety in the GP molecule. In addition, Persson et al. [43] suggested a dominant **M** motif for monomethyl phosphate (MMP; ${\mathrm{CH}}_{3}$–$H_{2}{\mathrm{PO}}_{4}$) adsorption onto goethite. However, similar to our previous studies [47,48], we find that GP forms stable and strong **B** or **BB** motifs, see Figure 2a. In addition, Lü et al. [80] suggested that molecules with similar binding mechanism like GP such as glucose 6-phosphate and adenosine mono/triphosphates form nonprotonated bidentate complexes within first ten minutes after mixing. Furthermore, OP is known to form **M**, **B**, and **BB** motifs when interacting with goethite [33,44,50,58]. This raises the question of why the **M** motif could be the dominant species for GP on goethite, despite the fact that the binding energy for the **B** motif is stronger, see Figure 2a. Note that the interaction energies presented in Figure 2a are per bond. For IHP, we found a transformation of **B** motif to **M** motif on diaspore surface [47,48] because of intermolecular HBs in IHP and its strong interaction with water. Based on this, and on the study by Li et al. [29], one might assume that GP’s HBs with water or its steric hinderance might hinder the **B** motif. Nevertheless, Abdala et al. [58] showed that at low (1.25 μmolm${}^{-2}$), medium (2.5 μmolm${}^{-2}$), and high (10 μmolm${}^{-2}$) surface loading of OP onto goethite, the observed motifs are at low **B** (48%), **BB** (47%), and **M** (≈0); at medium **B** (77%), **BB** (25%), and **M** (≈0); and at high **B** (≈0), **BB** (18%), and **M** (77%). This clearly shows that for low surface loading the **B** and **BB** motifs are dominant motifs whereas for high surface loading the **B** motif is not formed in the first place. In addition to surface loading, the surface type is also vital in deciding the dominant motifs of phosphates on surface [15,44]. For the 010 goethite surface, the theoretical study of single phosphate molecule interaction at goethite–water interface by Ahmed et al. [15], Ahmed et al. [46] showed that the **B** motif has a stronger binding than **M**. Further, the protonation state of GP’s phosphate group here could be understood from the study of Sheals et al. [81] of glyphosate (GLP, $H_{2}{\mathrm{PO}}_{3}$–${\mathrm{CH}}_{2}$–${\mathrm{NH}}_{2}$–${\mathrm{CH}}_{2}$–COO) on goethite; the twice deprotonated phosphate species dominate at the low surface adsorption densities and the monoprotonated phosphate species dominate at high surface adsorption densities. Correlating the current study with the above literature, the GP binds through a **B** motif with a twice deprotonated phosphate group dominantly at low surface adsorption density. At high surface adsorption density, GP binds preferentially through a **M** motif with a monoprotonated phosphate group.

The Fe–P distance observed here for the GP **M** motif is within the range of Fe–P distances observed in literature for goethite–phosphate complexes: 3.13–3.37 [82], 3.17–3.32 [83], 3.38 [44], 3.20 [46], and 3.41 [15], see Figure 2b. For the **B** motif, even though the calculated phosphate bond in this motif exhibits a strong interaction energy [46,47,48], it is not often discussed in experimental studies [22,44]. X-ray absorption studies of Abdala et al. [58] showed that the Fe–P distance observed for **B** motif case is around 2.85 Å, which is close to the value observed here, see Figure 2b. Furthermore, similar Fe–P values are observed for **B** motif in theoretical studies, i.e., 2.64 [46] and 2.58 Å [15]. Comparing the present interaction energies with other systems we notice the following. The present **B** motif exhibits stronger binding energy than the **M** motif, similar to the OP [46] and GLP [15] interaction with 010 goethite surface. Moreover, OP and GLP bind more strongly with goethite than with water, as observed for GP and water here. This suggests that phosphates often have the ability to replace water from goethite surface.

### 3.2. Goethite–IHP–Water Interaction

#### 3.2.1. IHP **M** Motifs

Here, the initial **M** (Fe–$O_{p}$) and **B** ($O_{p}$–Fe–$O_{p}$) motifs (see Appendix A) resulted in two different **M** motifs. **M(1)** corresponds to the initial **M** configuration, with an average Fe–$O_{p}$ bond length and Fe–P distance of 2.09 and 3.45 Å, respectively, see Figure 2b. Two proton transfer events are observed from IHP to water in the equilibration stage, see Figure 4a,b. Moreover, a proton transfer is observed from goethite to water, see Figure 4a. During the production trajectory, IHP is twice deprotonated and the protonation state of IHP has not altered. It has formed an average of 22 HBs with water. In addition, intramolecular HBs are observed between adjacent phosphate groups along the whole MD trajectory, see Figure 4b–d. Similar to our previous studies [47,48], the HBs formed between IHP and water and the intramolecular HBs in IHP are mostly strong to moderately strong HBs. Figure 4d shows the **M(1)** motif at 25 ps. The average goethite–IHP and IHP–water interaction energies observed here are −56 and −5.5 kcal/mol, respectively. The goethite-water interaction energy here is −5.7 kcal/mol which is less than in the GP’s **M** motif.

In **M(2)** motif case, the initial configuration was the **B** motif (see Appendix A), which transformed to form **M(2)**. Here, one of the Fe–$O_{p}$ bonds is dissociated, and for the other bond the average Fe–$O_{p}$ bond length and Fe–P distance is found to be 1.95 and 3.2 Å, respectively, see Figure 2b. In the equilibration phase, a proton transfer event is observed from IHP’s oxygen to the surface (see Figure 4a), followed by formation of an intramolecular HB between the same oxygen and an adjacent phosphate group, see Figure 4b. IHP has formed around 20 HBs with water and two of them are apparently strong enough such that proton transfer events are observed from IHP to water. In addition, IHP has formed multiple intramolecular HBs between phosphate groups, see Figure 4f–h. During the production trajectory, IHP is three times deprotonated with one proton transfer to surface and two to water. One of the protons transferred from IHP to water formed an HB with the goethite surface, see Figure 4g–h. The time averaged goethite–IHP interaction energy observed here is −190 kcal/mol which is higher than the **M(1)** motif case due to additional proton transfers from IHP to goethite surface observed here, see Figure 2e–h.

#### 3.2.2. IHP **3M** Motif

Here, IHP is aligned parallel to the surface to form three Fe–$O_{p}$ covalent bonds as shown in Figure 1c. A stable **3M** motif is observed during the production trajectory with mean average Fe–$O_{p}$ bond length of 1.96 Å and the corresponding mean average Fe–P distance of 3.2 Å, see Figure 2b, respectively. During equilibration, IHP has formed an average of 22 HBs with water of which three transformed to proton transfers from IHP to water, see Figure 4i,k. Furthermore, one proton transfer is observed from IHP to the surface, see Figure 4j. The equilibrated IHP is four times deprotonated with three proton transfers to water and one to the surface. The average number of HBs formed between IHP and water in production trajectory is around 23. The Figure 4l shows the IHP **3M** motif at 25 ps.

The time averaged goethite–IHP interaction energy per bond observed here is −86 kcal/mol. The goethite–IHP per bond interaction energy here is small compared to the **M(2)** motif case, but the overall interaction energy is higher than for both **M** motif cases, see Figure 2a. The IHP–water interaction energy is −7.8 kcal/mol which is higher than for the IHP **M** motif cases because of an additional proton transfer from IHP to water.

#### 3.2.3. Discussion of Goethite–IHP–Water Interactions

Evaluating the interaction of the IHP phosphate groups with goethite, the **B** motif is found to be unstable here, similar to our previous studies [47,48] for IHP interaction with diaspore. To our knowledge, studies in literature have not mentioned the existence of **B** or **BB** motifs for IHP interacting with minerals. However, the **M** motif is often observed in theoretical studies [45,47,48] for IHP. Interestingly, for the number of phosphate groups which will bind to the surface different suggestions exist, ranging from four [40], three [45,47], two [47,48], one [47,48], and 0 [30]. The present **M** and **3M** stable motifs add to the list. The goethite–water interaction energy observed here is less than in the GP case. The weaker interaction of water in the latter case is in line with the study by Celi et al. [31] who showed that IHP adsorption on goethite makes the surface more negative than in case of OP.

Numerous studies show that IHP has the ability to compete with OP for the same binding sites and it could release and replace OP on goethite [23]. Comparing IHP and GP interaction energies, IHP exhibits stronger adsorption onto goethite than GP (see Figure 2a) which is in line with experiment [84] and our previous studies [47,48] for diaspore. Therefore, IHP has the ability to replace GP and the order of adsorption energies is IHP > GP > water. The Fe–P distances observed here are in close match to the values observed in experimental [58,82] and theoretical studies [15,44,46,83] of OP adsorption onto goethite. This is in line with Celi et al. [85] suggestion that IHP phosphate groups react with goethite in the same way as OP. Compared to GP, one could also infer that the changes in the phosphate group’s geometry observed upon binding to goethite is similar for IHP and GP.

### 3.3. Theoretical IR Spectra of GP and IHP Adsorbed onto Goethite

The IR spectra of IHP/GP adsorbed onto goethite are calculated only for the stable motifs observed above, i.e, **M** and **B** of GP and **M** and **3M** of IHP. These motifs remained stable also during the extended trajectory used to calculate spectra, see Appendix A. The theoretical spectra calculated for these motifs are normalized and illustrated in Figure 5. The spectral analysis here is confined to 950–1250 ${\mathrm{cm}}^{-1}$ mainly to focus on the [P–O] stretching modes which are dominant in this range [22,86,87]. Therefore, the discussion below is mainly focused on phosphate ion bondings observed in the production trajectory and the corresponding IR spectra from literature. However, the peak positions of the IR spectra are known to shift differently with environmental factors such as water content and pH [50]. Especially nonprotonated oxygens of phosphate group are more sensitive to water hydrogens and hydroxyl groups and form HBs which might influence the spectra [50,88,89]. Both IHP and GP form strong to moderately strong HBs with surrounding water [47,48] and also contain nonprotonated oxygens, see Figure 3 and Figure 4. Therefore, the calculated spectra here should reflect these interactions. The charge superscript is ignored when representing the phosphate ion bondings, e.g., ${\mathrm{FeH}}_{2}{\mathrm{PO}}_{4}$ or ${\mathrm{FeHPO}}_{4}$, as these complexes are part of GP or IHP but not a molecule on their own.

Arai and Sparks [90] predicted the type of aqueous phosphate species based on the number of bands in IR spectra and the corresponding symmetry. According to their study, the aqueous phosphate species and their corresponding symmetry with number of bands (in parentheses) are ${\mathrm{PO}}_{4}$–$T_{d}\left(1\right)$, ${\mathrm{HPO}}_{4}$–$C_{3v}\left(3\right)$, $H_{2}{\mathrm{PO}}_{4}$–$C_{2v}\left(4\right)$, and $H_{3}{\mathrm{PO}}_{4}$–$C_{3v}\left(3\right)$. The study also showed that after OP binds to goethite the symmetry is reduced and the aqueous phosphate IR bands further split to form additional bands. Here, the symmetry of the phosphate group/s of IHP and GP is further reduced compared to OP bonded to goethite because of the P–O–C connection. Therefore, in the current study the spectrum assignment is performed based on the phosphate group moiety and its binding motif with goethite rather than symmetry.

Before we start the analysis some limitations about this approach should be mentioned. In an experimental set-up the IR spectra are obtained for an ensemble of molecules interacting with different goethite surface types. In contrast, here the analysis is performed for a single IHP/GP molecule interacting with only one goethite surface type. Furthermore, the IR bands and their corresponding assignments may vary based on pH, experimental conditions, and surface crystallinity [44]. Ahmed et al. [15] provided a novel approach to this problem by calculating weighted averages of theoretical IR spectra involving simultaneously the most common and abundant motifs and surface planes. This study also emphasized the idea that an abundant motif’s characteristics dominate the overall spectra which is in line with experimental studies [22,57].

#### 3.3.1. GP IR Spectra

In the GP **M** motif case, a deprotonated phosphate species is observed throughout the production trajectory. Therefore, the phosphate species observed here is ${\mathrm{FePO}}_{4}$. The IR spectrum for GP exhibits ten bands at 975, 998, 1029, 1042, 1070, 1120, 1146, 1176, 1211, and 1244 ${\mathrm{cm}}^{-1}$ in the frequency range 950–1250 ${\mathrm{cm}}^{-1}$, see Figure 5a. The first six peaks are of high to moderately high intense, whereas the remaining ones are less intense. The peaks at 1176 and 1211 ${\mathrm{cm}}^{-1}$ are less intense and wide ranged peaks distributed in frequency range of 1165 to 1187 ${\mathrm{cm}}^{-1}$ and 1200 to 1224 ${\mathrm{cm}}^{-1}$. In the GP **B** motif case also a deprotonated phosphate species is also observed throughout the simulation trajectory. Here, the GP exhibits ten bands at 951, 994, 1043, 1088, 1115, 1148, 1170, 1191, 1213, and 1236 ${\mathrm{cm}}^{-1}$. Here, only the first four bands are of high to moderately high intense and the remaining ones are less intense, see Figure 5a. The bands at 1115 ${\mathrm{cm}}^{-1}$ and 1170 ${\mathrm{cm}}^{-1}$ are shoulder bands distributed in the range of 1109 to 1119 ${\mathrm{cm}}^{-1}$ and 1164 to 1175 ${\mathrm{cm}}^{-1}$, respectively. The GP **M** and **B** motif cases have matching bands within $\pm 6\;{\mathrm{cm}}^{-1}$ of frequencies at 996, 1041, 1117, 1147, 1173, 1212, and 1240 ${\mathrm{cm}}^{-1}$. In the following we compare these findings with previous studies.

GP **M** motif case: Lincoln and Stranks [91] studied phosphate binding to Co in the $\mathrm{Co}{\left({\mathrm{NH}}_{3}\right)}_{5}{\mathrm{PO}}_{4}$ complex and assigned bands at 980 and 1030 ${\mathrm{cm}}^{-1}$ to [P–OCo] and [P–O] modes of monodentate deprotonated phosphate group (${\mathrm{CoPO}}_{4}$), respectively. Tribe et al. [83] assigned bands at 979 and 1032 ${\mathrm{cm}}^{-1}$ to [P–O] modes of ${\mathrm{FePO}}_{4}$ phosphate species for glyphosate adsorption onto goethite. Therefore, the bands at 975 and 1029 ${\mathrm{cm}}^{-1}$ here are assigned to [P–OFe] and [P–O] modes. Compared to the study by Li et al. [29] of GP adsorption onto goethite, the bands at 998, 1042, 1120, and 1146 ${\mathrm{cm}}^{-1}$ could be assigned to [P–O] stretching mode, see Table 1 and Figure 5. The band at 998 ${\mathrm{cm}}^{-1}$ could also be assigned to [P–OFe] stretching mode as multiple studies [15,22,57,83] assigned bands around this frequency to the same stretching mode. Tribe et al. [83] assigned a band at 1063 ${\mathrm{cm}}^{-1}$ to [P–O] stretching mode for the ${\mathrm{FeHPO}}_{4}$ complex, and therefore the band at 1070 ${\mathrm{cm}}^{-1}$ here could be assigned to a [P–O] stretching mode. The less intense band at 1176 ${\mathrm{cm}}^{-1}$ might correspond to [P–O] asymmetric stretching mode as Li et al. [29] assigned a band at 1180 ${\mathrm{cm}}^{-1}$ to [P–O] or [P–OH] for ${\mathrm{HPO}}_{4}$ phosphate species of aqueous GP. Moreover, Persson et al. [57] assigned a band at 1178 ${\mathrm{cm}}^{-1}$ to [P–O] stretching mode for ${\mathrm{FeH}}_{2}{\mathrm{PO}}_{4}$ and Kubicki et al. [44] listed a band at 1176 ${\mathrm{cm}}^{-1}$ for the ${\mathrm{FeHPO}}_{4}$ phosphate species. The band at 1211 ${\mathrm{cm}}^{-1}$ could be assigned to stretching modes of [C–C–C] and [C–O–C] of glycerol group based on the assignment by Nakagawa and Oyama [92] of 1210 ${\mathrm{cm}}^{-1}$ frequency observed in glycerol–water interactions [93]. Regarding the next band at 1244 ${\mathrm{cm}}^{-1}$, Kubicki et al. [44] assigned the band at 1250 ${\mathrm{cm}}^{-1}$ to [P–O–H] bending vibrations and in addition, Ahmed et al. [15] assigned the band at 1227 ${\mathrm{cm}}^{-1}$ to [P–O–H] bending vibrations for OP on 010 goethite surface. Therefore, the band at 1244 ${\mathrm{cm}}^{-1}$ here could be assigned to GP’s unprotonated phosphate oxygens interaction with water hydrogens.

GP **B** motif case: Ahmed et al. [15] study of OP on 010 goethite surface assigned bands at 957 and 964 ${\mathrm{cm}}^{-1}$ to [O–P–2OFe] and [2O–P–OFe] symmetric stretching modes and Persson et al. [57] assigned the band at 966 ${\mathrm{cm}}^{-1}$ to [P–O] stretching mode. Therefore, the band at 951 ${\mathrm{cm}}^{-1}$ could be assigned to [P–O] or [P–OFe] stretching modes. The band at 994 ${\mathrm{cm}}^{-1}$ frequency is close to 998 ${\mathrm{cm}}^{-1}$ frequency found in GP **M** motif case, and therefore it is assigned the same modes, [P–O] or [P–OFe] stretching modes, as the latter frequency. The bands at 1043, 1115, and 1148 ${\mathrm{cm}}^{-1}$ could be assigned to [P–O] stretching mode as per Li et al. [29] study while the band at 1088 ${\mathrm{cm}}^{-1}$ could be assigned to [P=O] [29] or [P–O] [83] stretching modes, see Table 1. The band at 1170 ${\mathrm{cm}}^{-1}$ is assigned [P–O] mode which is the same mode assigned for the band at 1176 ${\mathrm{cm}}^{-1}$ in GP **M** motif case. The band at 1191 and 1213 ${\mathrm{cm}}^{-1}$ could be assigned to [C–C–C] and [C–O–C] stretching modes of glycerol group as assigned for 1211 ${\mathrm{cm}}^{-1}$ frequency observed in GP **M** motif case. The band at 1236 ${\mathrm{cm}}^{-1}$ is assigned the same mode as 1244 ${\mathrm{cm}}^{-1}$ in GP **M** motif case.

Comparing IR spectra of GP **M** and **B** motifs, we conclude that several common band assignments are observed between both cases. The bands between 951 and 998 ${\mathrm{cm}}^{-1}$ are assigned to [P–O] or [P–OFe] stretching modes, and the ones between 1029 and 1176 ${\mathrm{cm}}^{-1}$ are assigned to [P–O] stretching mode except for 1088 ${\mathrm{cm}}^{-1}$ frequency which could be assigned to [P=O] stretching mode as well. The bands between 1191 and 1213 ${\mathrm{cm}}^{-1}$ are assigned to [C–C–C] and [C–O–C] stretching modes of glycerol group, respectively, while the bands between 1236 and 1244 ${\mathrm{cm}}^{-1}$ are assigned to unprotonated phosphate oxygens interaction with water hydrogens. The comparison of spectra calculated here with spectra from the experimental study by Li et al. [29] is shown in Appendix A. Moreover, spectral peaks characteristic to water are analyzed in light of peaks from literature, see Appendix A.

#### 3.3.2. IHP IR Spectra

IHP has six phosphate groups and hence the theoretical spectra calculated here would include the characteristics of different types of phosphate species originating from the six phosphate groups. In addition, intermolecular HBs are also common between adjacent phosphate groups which shift or alter intensity of the IR spectra [88,89]. In the IHP **M** case, the six phosphate groups transformed to two types of phosphate species, ${\mathrm{HPO}}_{4}$ (2) and $H_{2}{\mathrm{PO}}_{4}$ (4), in the production trajectory. The IR spectra for IHP **M** case exhibited bands at 970, 991, 1036, 1063, 1090, 1148, 1190, 1214, and 1236 ${\mathrm{cm}}^{-1}$, see Figure 5b. The first five bands are high intense bands while the latter ones are moderately intense to often less intense. The band at 1090 ${\mathrm{cm}}^{-1}$ is a shoulder band distributed in the frequency range of 1084 to 1099 ${\mathrm{cm}}^{-1}$ while the band at 1214 is a least intense flat band distributed in the frequency range of 1205 to 1222 ${\mathrm{cm}}^{-1}$. In IHP **3M** motif case all possible phosphate species, ${\mathrm{PO}}_{4}$ (1), ${\mathrm{HPO}}_{4}$ (2), $H_{2}{\mathrm{PO}}_{4}$ (3), are observed in the production trajectory. The IR spectra have bands at 958, 996, 1023, 1050, 1065, 1096, 1108, 1141, 1164, 1193, and 1219 ${\mathrm{cm}}^{-1}$. The bands at 1065 and 1219 ${\mathrm{cm}}^{-1}$ are shoulder bands distributed in the frequency range of 1059 to 1071 ${\mathrm{cm}}^{-1}$ and 1212–1225 ${\mathrm{cm}}^{-1}$, respectively. The bands at 1096 and 1108 ${\mathrm{cm}}^{-1}$ are maxima of a bimodal peak which is distributed in the frequency range of 1084 to 1116 ${\mathrm{cm}}^{-1}$. The IHP **M** and IHP **3M** motif cases have matching bands within $\pm 6\;{\mathrm{cm}}^{-1}$ of frequencies at 964, 993, 1064, 1093, 1144, 1191, and 1217 ${\mathrm{cm}}^{-1}$. The IHP spectra obtained here are deconvoluted into Gaussian functions using the PeakFit v4 software [95] to compare current results with Johnson et al. [30] study. PeakFit v4 is the same software used by Johnson et al. [30] for deconvolution of IHP spectra. The deconvoluted spectra and the corresponding fitted band centers are presented in Appendix A.

IHP **M** case: Guan et al. [45] assigned bands between 970–985 ${\mathrm{cm}}^{-1}$ and 970 ${\mathrm{cm}}^{-1}$ frequency to [P–O–C] stretching mode for IHP adsorbed onto aluminum hydroxide and for IHP in water, respectively. In contrast, Johnson et al. [30] assigned a fitted band center at 974 ${\mathrm{cm}}^{-1}$ to [P–O] mode but he assigned fitted band center at 991 and 1011 ${\mathrm{cm}}^{-1}$ to [P–O–C] stretching mode. Therefore, the peaks at 970 and 991 ${\mathrm{cm}}^{-1}$ (fitted band centers at 967 and 992 ${\mathrm{cm}}^{-1}$, respectively, see Appendix A) could be assigned to [P–O] or [P–O–C] stretching modes. The band at 1036 ${\mathrm{cm}}^{-1}$ (fitted band center 1036 ${\mathrm{cm}}^{-1}$) might correspond to the intermolecular HBs between adjacent phosphate groups or between a phosphate group and water adsorbed to goethite surface. This assignment is based on the study of Johnson et al. [30], where the fitted band center at 1043 ${\mathrm{cm}}^{-1}$ is assigned to P–O⋯H (H from the phosphate group or water adsorbed to goethite surface). The high intense peak at 1063 ${\mathrm{cm}}^{-1}$ with fitted band center at peak center could be assigned to [P–O] stretching mode of ${\mathrm{HPO}}_{4}$ as per Johnson et al. [30] and Yan et al. [41] studies, see Table 1. The next peak is at 1090 ${\mathrm{cm}}^{-1}$ (fitted band center at 1091 ${\mathrm{cm}}^{-1}$) and it could be assigned to [P–O] mode as Johnson et al. [30] assigned a fitted band at 1099 to such a mode. The fitted band center at 1120 ${\mathrm{cm}}^{-1}$ which does not have a specific assigned peak (see Appendix A) might correspond to [P–O] stretching mode as Johnson et al. [30] assigned fitted band center at 1128 ${\mathrm{cm}}^{-1}$ to [P–O] stretching mode. The peak at 1148 ${\mathrm{cm}}^{-1}$ (fitted band centers at 1147 and 1165 ${\mathrm{cm}}^{-1}$) might correspond to [P–OFe] stretching mode as Guan et al. [45] and Yan et al. [41] assigned bands 1148 ${\mathrm{cm}}^{-1}$, 1157 ${\mathrm{cm}}^{-1}$, and 1166 ${\mathrm{cm}}^{-1}$ to [P–OAl] and [P–OFe] stretching modes, respectively. It could also be assigned to [P–O] mode as Johnson et al. [30] assigned 1160 ${\mathrm{cm}}^{-1}$ band center to such a mode. The peak at 1190 ${\mathrm{cm}}^{-1}$ the fitted band center is at peak center and it is assigned to [P–O] mode as per Johnson et al. [30]. The band at 1214 ${\mathrm{cm}}^{-1}$ (fitted band center at 1213 ${\mathrm{cm}}^{-1}$) could be assigned to [P=O] stretching mode. This is because, Celi et al. [31] suggested that after IHP adsorption onto goethite the [P=O] stretching band observed for aqueous IHP at 1223 ${\mathrm{cm}}^{-1}$ frequency might shift to lower frequencies. In addition, Johnson et al. [30] assigned the fitted band center at 1220 ${\mathrm{cm}}^{-1}$ to [P=O] stretching mode. Bands within 1230 and 1250 ${\mathrm{cm}}^{-1}$ are not common in goethite–IHP-related studies, see Table 1, and therefore the band at 1239 ${\mathrm{cm}}^{-1}$ is assigned to [P–O–H] bending vibrations based on Ahmed et al. [15]’s study.

IHP **3M** motif case: The first band at 958 ${\mathrm{cm}}^{-1}$ could be assigned to [P–O] or [P–O–C] stretching modes based on reasons for assignment of 970 ${\mathrm{cm}}^{-1}$ band in IHP **M** motif case. The next band at 996 ${\mathrm{cm}}^{-1}$ is assigned to [P–O] or [P–O–C] stretching modes for the same reasons as given for the assignment of the 977 ${\mathrm{cm}}^{-1}$ band in IHP **M** motif case. The band at 1023 ${\mathrm{cm}}^{-1}$ (fitted band center at 1022 ${\mathrm{cm}}^{-1}$) could be assigned to [P–O–C] stretching mode, as Johnson et al. [30] assigned the fitted band center at 1011 ${\mathrm{cm}}^{-1}$ to such a mode. Similar to the band at 1036 ${\mathrm{cm}}^{-1}$ for IHP **M** motif case, the band at 1050 ${\mathrm{cm}}^{-1}$ (fitted band center at 1048 ${\mathrm{cm}}^{-1}$) here could be assigned to inter- and intramolecular HBs among adjacent phosphate groups and a phosphate group with water bound to goethite, respectively. The next bands at 1065 and 1096 ${\mathrm{cm}}^{-1}$ (corresponding fitted band centers at 1068 and 1092 ${\mathrm{cm}}^{-1}$) are assigned to [P–O] stretching modes similar to peaks at 1063 and 1090 ${\mathrm{cm}}^{-1}$ of IHP **M** motif case, see Table 1. The peak at 1108 ${\mathrm{cm}}^{-1}$ with fitted band center at 1113 ${\mathrm{cm}}^{-1}$ could be assigned to [P–O] stretching mode based on Johnson et al. [30] assignment of fitted band center at 1127 ${\mathrm{cm}}^{-1}$ to [P–O] stretching mode. The band at 1141 ${\mathrm{cm}}^{-1}$ (fitted band center at 1140 ${\mathrm{cm}}^{-1}$) is assigned to [P–OFe] stretching mode as Guan et al. [45] assigned the band at 1048 ${\mathrm{cm}}^{-1}$ to [P–OAl] stretching mode. It could also be assigned to [P–O] mode since Yan et al. [41] assigned the band at 1135${\mathrm{cm}}^{-1}$ to [P–O] mode. The band 1164 ${\mathrm{cm}}^{-1}$ (fitted band center at 1165${\mathrm{cm}}^{-1}$) is assigned to [P–OFe] or [P–O] mode, as Yan et al. [41] assigned the band at 1166 ${\mathrm{cm}}^{-1}$ to [P–OFe] stretching mode and Johnson et al. [30] assigned fixed band center at 1162 to [P–O] mode. The peaks at 1193 and 1219 ${\mathrm{cm}}^{-1}$ are assigned to [P–O] and [P=O] modes for the same reasons given the assignment of peaks at 1190 and 1214 ${\mathrm{cm}}^{-1}$ in IHP **M** motif case.

Summarizing the IHP **M** and **3M** motifs’ IR spectra, and their assignments based on literature (see Table 1), we conclude the following. The bands in the frequency range of 958 to 996 ${\mathrm{cm}}^{-1}$ could be mainly assigned to [P–O] or [P–O–C] stretching modes. The bands around 1043 ${\mathrm{cm}}^{-1}$ are assigned to intermolecular HBs between IHPs phosphate groups and intramolecular HBs between IHP phosphate groups and water adsorbed to goethite. The bands observed in the range of 1063 to 1108 ${\mathrm{cm}}^{-1}$ are assigned to [P–O] stretching modes and the ones in 1141–1164 ${\mathrm{cm}}^{-1}$ frequency range are assigned to [P–OFe] or [P–O] modes. The peaks around 1191 ${\mathrm{cm}}^{-1}$ are assigned to [P–O] modes and the ones around 1217 ${\mathrm{cm}}^{-1}$ are assigned to [P=O] modes. The comparison of IHP spectra with spectra from the experimental study of Yan et al. [41] is shown in Appendix A. A few selected frequency ranges where the IR spectra of water could be characterized are analyzed, see Appendix A.

## 4. Summary and Conclusions

The world population is expected to increase 34% by 2050 and will reach 9.1 billion [96], challenging the agricultural industry to meet the nutritional needs. Unfortunately, agricultural production is heavily dependent on the soon to be exhausted P rocks. Therefore, there is a need to find ways to recycle and secure P resources to support the raising global population. The current study focused on P interaction with soil minerals, a significant factor that causes P inefficiency and P loss in soil. The present results are expected to provide additional insight into organic P interaction at goethite–water interface.

Analysis of binding energies (see Figure 2) show that the GP **B** motif exhibits stronger overall binding energy than the **M** motif. This is in contrast to the suggestion by Li et al. [29] that GP might not form **B** motif due to steric hinderance of organic moiety. However, based on interaction energies calculated here and the study by Abdala et al. [58], we conclude that GP forms a **B** motif at low goethite surface loading, while at high loading the **M** motif dominates.

Regarding IHP, it is found to interact with goethite through multiple phosphate groups and its **3M** motif has a higher interaction energy than the **M** motif. Furthermore, its **B** motif is unstable which transformed to the **M** motif. The transformation of the **B** to the **M** motif has occurred in our previous studies [47,48] as well, confirming that IHP phosphate groups are more likely to form **M** motifs with goethite. The IHP interaction energy with goethite is stronger than for GP which suggests that it could replace GP to bind with goethite. This is in line with the results of De Groot and Golterman [97] which showed that IHP can replace OP and inhibit it from further adsorption.

The energy required for phosphate to replace a ${\mathrm{OH}}^{-}$ from the surface could not be estimated from that study. However, Ahmed et al. [15] showed that ${\mathrm{OH}}^{-}$ has a higher interaction energy than OP and water and thus could replace both. However, in contrast, Li et al. [29] showed that GP replaces ${\mathrm{OH}}^{-}$ at the goethite surface at a high rate. Therefore, understanding the priority of adsorption is important as a non-replaceable hydroxyl group at the surface could restrict the IHP interaction with goethite and also reduce the range of binding motifs. The current study proposes a competitive adsorption study for multiple instances of IHP/GP in presence of water and ${\mathrm{OH}}^{-}$ at the goethite surface to understand effect of high surface loading and influence of ${\mathrm{OH}}^{-}$ on binding motifs.

Finally, we have investigated the IR signatures of the different binding motifs. The IR spectra calculated for GP **M** and **B** motifs match reasonably with phosphates related IR spectra from literature. This suggests that both motifs might exist simultaneously on the goethite surface. However, given the similarity of spectra, the proportion of these motifs could not be estimated as done by Ahmed et al. [15]. The calculated IR spectra for IHP also matches well with the IR spectra from literature. This validates the current modeling approach for simulating the interactions of IHP and GP at goethite–water interface.

## Figures and Tables

**Figure 1 molecules-26-00160-f001:**
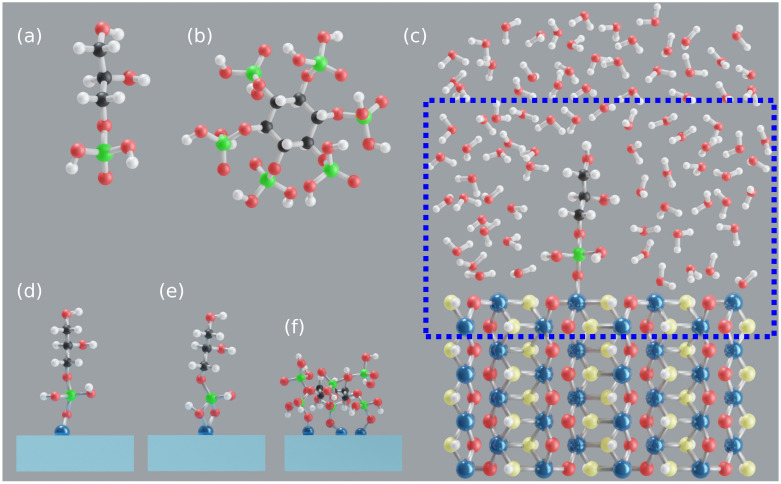
GP [$C_{3}H_{9}O_{6}P$] (**a**), IHP [$C_{6}H_{18}O_{24}P_{6}$] (**b**), goethite-GP-water complex (**c**), **M** motif (**d**), **B** motif (**e**), and **3M** motif (**f**). Blue, red, yellow, white, black, and green colors correspond to iron, bridging oxygen, hydroxyl oxygen, hydrogen, carbon, and phosphorus atoms, respectively. The dotted line denotes the QM box.

**Figure 2 molecules-26-00160-f002:**
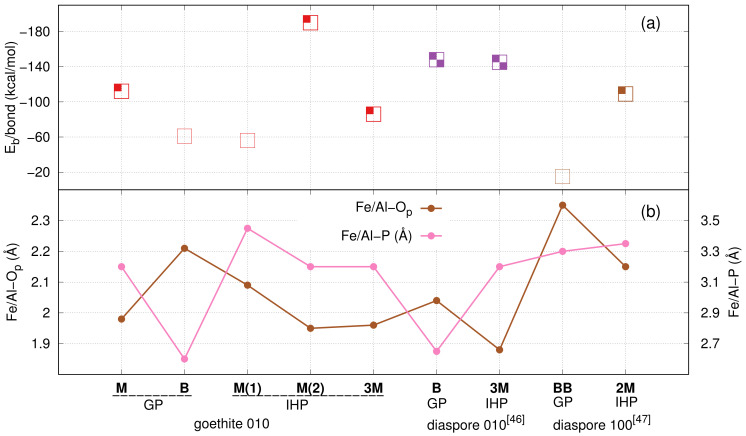
Comparison plot of interaction energies per bond $E_{b}$ (**a**) and Fe–$O_{p}$, Fe–P distances (**b**). Panels (**a**,**b**) also contain data for diaspore–IHP/GP–water complexes [47,48]. In the top figure, the number of filled subsquares denote the total number of proton transfers from IHP/GP to surface and center of box denotes the binding energy. The square with zero filled subsquares denotes zero proton transfers from phosphate to the surface.

**Figure 3 molecules-26-00160-f003:**
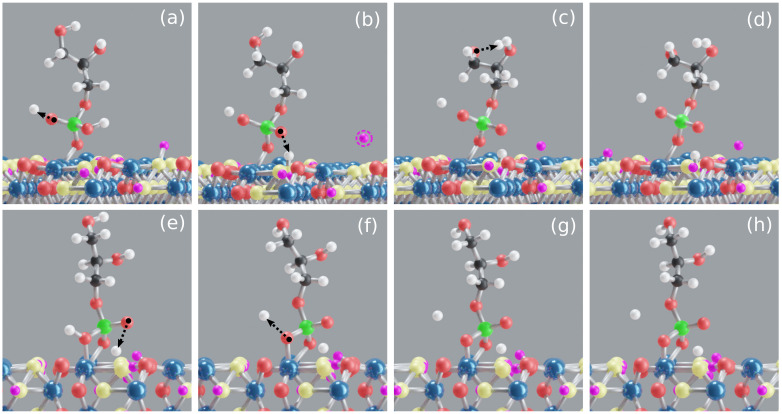
Snapshots of goethite–GP–water models along the simulation trajectory. Proton transfer from GP to water (**a**), a proton transfer from GP to surface and from surface to water (**b**), a proton transfer from GP to water (**c**), a stable **M** motif at 25 ps (**d**), proton transfers from GP to water (**e**,**f**), and **B** motif at 20 ps (**g**) and 25 ps (**h**). As proton transfer events are common in these interactions, the goethite surface hydrogen atoms and GP’s hydrogen atoms are shown in violet and white colors to avoid confusion. Moreover, the surrounding water is ignored here for better visualization. Snapshots (**a**,**b**,**e**,**f**) are from equilibration phase.

**Figure 4 molecules-26-00160-f004:**
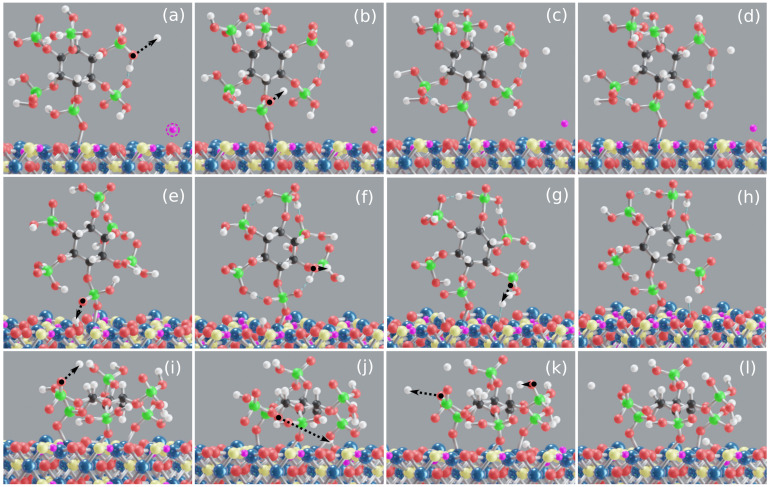
Snapshots of goethite–IHP–water models along the simulation trajectory. In **M(1)** motif, proton transfer from IHP to water and from goethite surface oxygen to water (**a**), intramolecular HB and a proton transfer from IHP to water (**b**), and **M(1)** motif at 17 and 25 ps (**c**,**d**). In **M(2)** motif, proton transfer from IHP to surface (**e**), protons transfer from IHP to water (**f**,**g**), and **M(2)** motif at 25 ps (**h**). In **3M** motif, three proton transfers from IHP to water (**i**,**k**) and a proton transfer from IHP to surface (**j**), **3M** motif at 25 ps (**l**). As proton transfer events are common in these interactions, the goethite surface hydrogen atoms and IHP’s hydrogen atoms are shown in violet and white colors to avoid confusion. Moreover, the surrounding water is ignored here for better visualization. Snapshots that show proton transfers are from the equilibration phase.

**Figure 5 molecules-26-00160-f005:**
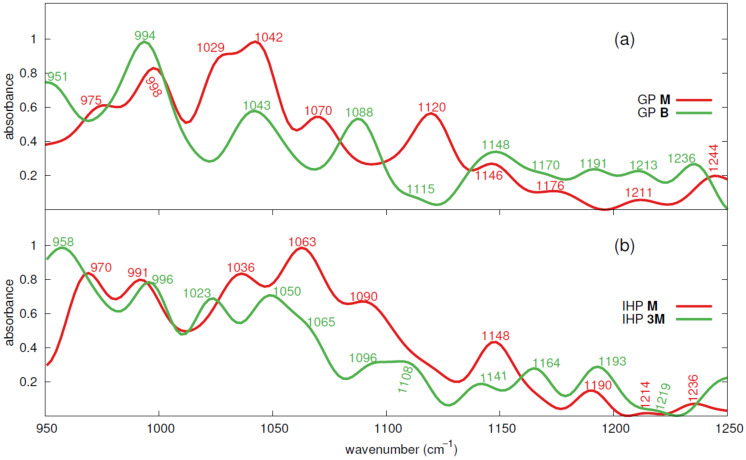
Calculated IR spectra of GP **M** and **B** motifs (**a**) and IHP **M** and **3M** motifs (**b**).

**Table 1 molecules-26-00160-t001:** Experimental and theoretical IR frequencies from selected studies related to phosphates OP, GP, IHP, and others on goethite are listed below. The IR frequencies calculated here for GP and IHP on goethite are also presented here. The symbols †, ‡, •, ∘, ▴, ⋯, and ⋆ denote [P–O], [P=O], [P–OFe], [P–OH], [P–O–C], [P–O⋯H], and [C–C–C] plus [C–O–C] modes, respectively. The symbols are assigned to frequencies only when specifically mentioned in reference experimental study. The frequency assigned to multiple modes is denoted as •/∘ represent [P–OFe] or [P–OH] modes while •&/∘ represent [P–OFe] and/or [P–OH] mode.

Study and Description	Wavenumber [cm${}^{-1}$]											
goethite–OP												
Ref.Tejedor and Anderson [22]												
pH 6.0, 190 μmol P/g of Gt		1004	1030	1045		1099		1128				
${\mathrm{Fe}}_{2}{\mathrm{HPO}}_{4}$	982${}^{•/\circ }$	1006 ${}^{•}$						1123 ${}^{‡}$				
${\mathrm{Fe}}_{2}{\mathrm{PO}}_{4}$				1044 ${}^{†}$		1096 ${}^{†}$						
FePO${}_{4}$		1001${}^{•\&/†}$	1025 ${}^{†}$									
Ref. Persson et al. [57]												
pH 4.2−5.7 [${\mathrm{FeH}}_{2}{\mathrm{PO}}_{4}$]		1001${}^{•/\circ }$		1049${}^{\circ }$				1122 ${}^{†}$	1178 ${}^{†}$			
pH 7.9 [${\mathrm{FeHPO}}_{4}$]		1001${}^{•/\circ }$		1049${}^{\circ }$				1122 ${}^{†}$				
pH 13 [${\mathrm{FePO}}_{4}$]	966 ${}^{†}$			1057 ${}^{†}$								
Ref. Kubicki et al. [44]												
pH 4.22 [${\mathrm{FeH}}_{2}{\mathrm{PO}}_{4}$]	982	1009		1044		1091	1122	1157		1195		
pH 7.51 [${\mathrm{FeHPO}}_{4}$]			1022	1043	1084		1124		1177			
Ref. Ahmed et al. [15]												
pH 6.3		1000					1110		1165			
**M**@010 [${\mathrm{FeHPO}}_{4}$]	964		1021	1049							1207	
**B**@010 [${\mathrm{FeHPO}}_{4}$]	957	992	1007	1035	1071						1227	
**M**@100 [${\mathrm{FePO}}_{4}$]	957	999		1044	1058							
**B**@100 [${\mathrm{Fe}}_{2}{\mathrm{HPO}}_{4}$]	953	974		1048								
goethite–[organic P-compounds]												
Ref. Persson et al. [43] (MMP)												
pH 2.6 ${\mathrm{FeHPO}}_{4}$		1003 ${}^{†}$		1051 ${}^{†}$				1140 ${}^{†}$		1185		
pH 9.9 ${\mathrm{FePO}}_{4}$		990 ${}^{†}$		1051 ${}^{†}$				1120 ${}^{†}$		1185		
Ref. Tribe et al. [83] (GLP)												
${\mathrm{FeHPO}}_{3}$		1001 ${}^{•}$	1020 ${}^{•}$	1030 ${}^{†}$	1063 ${}^{\circ }$		1118 ${}^{†}$					
${\mathrm{FePO}}_{3}$	973 ${}^{†}$	979 ${}^{†}$		1032 ${}^{†}$	1068 ${}^{†}$	1084 ${}^{†}$	1095 ${}^{†}$	1102 ${}^{†}$				
Ref. Li et al. [29] (GP)												
pH 3		1008 ${}^{†}$		1052 ${}^{†}$		1098 ${}^{‡}$		1139 ${}^{†}$				
pH 9		998 ${}^{†}$		1044 ${}^{†}$		1092 ${}^{‡}$		1126 ${}^{†}$				
current work												
**M** [${\mathrm{FePO}}_{4}$]	975 ${}^{†/•}$	998 ${}^{†/•}$	1029 ${}^{†}$	1042 ${}^{†}$	1070 ${}^{†}$		1120 ${}^{†}$	1146 ${}^{†}$	1176 ${}^{†}$		1211 ${}^{⋆}$	1244 ${}^{\circ }$
**B** [${\mathrm{FePO}}_{4}$]	951 ${}^{†/•}$	994 ${}^{†/•}$		1043 ${}^{†}$		1088 ${}^{†/‡}$	1115 ${}^{†}$	1148 ${}^{†}$	1170 ${}^{†}$	1191 ${}^{⋆}$	1213 ${}^{⋆}$	1236 ${}^{\circ }$
goethite–IHP												
Ref. Yan et al. [41]												
pH 5		998 ${}^{†}$			1075 ${}^{†}$			1135 ${}^{†}$	1166 ${}^{•}$			
Ref. Yan et al. [94]												
pH 5		1010			1071			1134	1164			
pH 6		998			1068			1131				
Ref. Johnson et al. [30]												
fitted band centers	974 ${}^{†}$	991 ${}^{▴}$	1011 ${}^{▴}$	1043 ${}^{\cdots }$	1074 ${}^{†}$	1099 ${}^{†}$		1128	1160 ${}^{†}$	1187 ${}^{†}$	1220 ${}^{‡}$	
current work												
**M**	970 ${}^{†/▴}$	991 ${}^{†/▴}$		1036 ${}^{\cdots }$	1063 ${}^{†}$	1090 ${}^{†}$			1148 ${}^{†/•}$	1190 ${}^{†}$	1214 ${}^{‡}$	1236 ${}^{\circ }$
**3M**	958 ${}^{†/▴}$	996 ${}^{†/▴}$	1023 ${}^{▴}$	1050 ${}^{\cdots }$	1065 ${}^{†}$	1096 ${}^{†}$	1108 ${}^{†}$	1141 ${}^{†/•}$	1164 ${}^{†/•}$	1193 ${}^{†}$	1219 ${}^{‡}$	

## Data Availability

Not applicable.

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
