# Peer review of "Ab Initio Molecular Dynamics Simulations of the Interaction between Organic Phosphates and Goethite"

_molecules, 2020, doi:10.3390/molecules26010160_

Round 1
Reviewer 1 Report
The authors presented a thorough computational study of interaction of glycerolphosphate and inositol hexaphosphate with mineral goethite in water media. The importance of this study is anchored on absorption of fertilizers into soil minerals which reduces their availability to plants. The manuscript is clearly written and I consider the reported findings significant enough to warrant the publication of the work in the Molecules
I recommend the publication in present form. However, some typos should be corrected (interation->interaction etc)
Author Response
Thanks so much to the reviewer for his opinion about clarity and significance of our work and his recommendation by publication of our manuscript.
Comments: I recommend the publication in present form. However, some typos should be corrected (interation->interaction etc)
We have checked and corrected all typos and spelling mistakes as suggested by the reviewer.
Reviewer 2 Report
The authors have performed the QM/MM simulations of Goethitic IHP/GP water complexes and have studied the interaction energy and IR spectra. The manuscript can be accepted after the inclusion of minor corrections as mentioned below:
Since the role of water is also being investigated, it is very imperative to check for the effect on the surface and ligands on the IR spectra of liquid water. This should be included in the main text or in the supplementary information.
Further, for the modes assigned in IR spectrum have been characterized but the algorithm is not described? Is it based on previous literature or experiments?
The theoretical formulation for the usage of Voronoi tesselation based IR spectrum as implemented in TRAVIS might be mathematically included.
The reference for CP2K should be updated :
https://doi.org/10.1063/5.0007045
Author Response
Thanks to the reviewer for his valuable comments that would improve quality of our manuscript and thanks for his recommendation by publication of our manuscript.
Comments:
Since the role of water is also being investigated, it is very imperative to check for the effect on the surface and ligands on the IR spectra of liquid water. This should be included in the main text or in the supplementary information.
ANSWER
Thanks for the reviewer for referring to this point. In the revised manuscript, we have considered this issue by analyzing the calculated IR spectra in light of the IR spectra of water from literature. The analysis is presented in the supplementary information, please see Figure S8 and its descriptive caption in page 5
---------------------
Further, for the modes assigned in IR spectrum have been characterized but the algorithm is not described? Is it based on previous literature or experiments?
ANSWER
The present IR spectra were assigned based on previous experimental and theoretical studies of goethite-IHP/GP-water complexes. This was already described in the main text but to make clear we provided related information from the previous cited studies in Table 1.
-----------------------
The theoretical formulation for the usage of Voronoi tesselation based IR spectrum as implemented in TRAVIS might be mathematically included.
ANSWER
We have used TRAVIS software and cited to the original work and the implemented algorithm in TRAVIS. However, we prefer to not repeat the theoretical formulation here since we have not any new additional input in this direction. Moreover, we would like to deliver a message to readers that the main target in the present manuscript is investigating and analyzing the phosphate binding at the goethite-water interface.
-----------------
The reference for CP2K should be updated: https://doi.org/10.1063/5.0007045
Thanks so much for providing the updated reference for CP2K. We have updated the reference accordingly as suggested.
Reviewer 3 Report
A more efficient and sustainable phosphorus cycle is a major challenge to secure food security for a growing population. In this context, this paper aim to better understand how phosphate fertilizers attache to soil particles (Goethite). To do so, they used QM/MM simulations to identify and quantify the surface interactions between two phosphate compounds (glycerolphosphate (GP) & inositol hexaphosphate (IHP)) and goethite surface and the role of water in this interaction.
Major comments:
Concerning the goethite-GP interaction (paragraph 3.1.), two GP motif were observed (Monodentate (M) and Bidentate (B) motifs). According to the simulation, the GP B motif is more stable than the M motif but experimentally the M is the dominant species (except at low concentration here the B motif dominate). Since, the GP M motifs relies on a intramolecular H transfer, one could suggest that similar H transfers between neighboring adsorbed-GP happens at high surface loading thus favoring the M motifs.
Concerning the goethite-IHP interaction (paragraph 3.3.), the B motif is unstable and only three M motifs are observed : M(1), M(2) and 3M. The author suggested that the IHP adsorption decrease the interaction between water molecules and goethite surface probably because of the large size of the IHP molecule. I find that this phenomenon lacks an explanation. How the size of the IHP molecule can impact the surrounding water? IHP exhibits 22 hydrogen bonds with water (three time more than for GP molecule), thus IHP appears to have a strong hydration (e.g. outer-sphere complexation) shell which could affect surrounding water and hinders IHP adsorption.
Minor comments:
Do the authors consider studying the effect of pH which seems to be an important parameter?
Did the authors observed a reorganization of the surface layers of goethite crystal?
I suggest to the authors to put the description of the IR spectrum in a table for easier reading.
p1 line 22 : “effect” → “affect” or “have an effect”
p2 line 35 “these colloidals” → “these colloids”
p2 line 37 “Orthophosphate (OP) is one of most...” → “Orthophosphate (OP) is one of the most...”
p2 line 52: “outersphere complexes.” → “outer-sphere complexes.”
p2 line 79 : “Some of the common goethite surface planes...” → “Some of the common goethite surface planes...”
p3 line 86 : “pH less than the goethite PZC” → “pH lower than the goethite PZC”
p3 line 94 : “covalent bond/s” → “covalent bonds”
p4 line 116 : “they should allow to draw conclusions” → “they should allow us to draw conclusions”
p4 line 143 : “that differ based on level of theory” → “that differ based on the level of theory”
p5 line 164 : “( ≈ 2TB)” → “( ≈ 2 TB)”
p6 line 172, 193 and 195: “equlibration” → “equilibration”
p6 line 190: “and an Fe–P distance” → “and an Fe–P distance
p6 line 195 : ”trajecotry” → “trajectory”
p6 line 210: “Also, OP is know to form” → “Also, OP is known to form”
p6 line 211: “the question why” → “the question of why”
p7 line 216: “Neverthless” → “Nevertheless”
p7 line 231: “within range” → “within the range”
p8 line 247: “Also a proton transfer” → “Also, a proton transfer”
p9 line 269: “to surface” → “to the surface”
p9 line 273: . “Also one proton transfer” → “Also, one proton transfer”
p9 line 290: “in latter case” → “in the latter case”
p9 line 292: “covers larger part” → “covers a larger part”
p9 line 293 : “Consquently” → “Consequently“
p9 line 301: “one could also infer that that the changes...” → “one could also infer that that the changes...”
p10 line 320: “(in parenthesis)” → “(in parentheses)”
p11 line 366: “Regarding next band” → “Regarding next the band”
p11 line 358, 373, 376, 377 and many other time in the mnsucript: “strectching” → “strectching”
p13 line 383: “betweens” → “betweens“
p13 line 368: “comparision” → “comparision”
p14 line 450, 451, 453: “assigned band” → “assigned the band”
p14 line 455: “for same reasons” → “for the same reasons”
p14 line 468: “The bands in frequency range” → “The bands in the frequency range”
p14 line 471: Space missing at the beginning of the sentence
p15 line 480: “Also its B motif” → “Also, its B motif”
p15 line 490: “at surface” → “at the surface”
p15 line 492: “at goethite surface” → “at the goethite surface”
p15 line 494: Space missing at the beginning of the sentence
Author Response
Concerning the goethite-GP interaction (paragraph 3.1.), two GP motif were observed (Monodentate (M) and Bidentate (B) motifs). According to the simulation, the GP B motif is more stable than the M motif but experimentally the M is the dominant species (except at low concentration here the B motif dominate). Since, the GP M motifs relies on a intramolecular H transfer, one could suggest that similar H transfers between neighboring adsorbed-GP happens at high surface loading thus favoring the M motifs.
ANSWER:
We have not observed intramolecular proton transfer in GP for both M and B motifs. Therefore, we assume that reviewer means intermolecular proton transfers.
Since our simulations have been performed in the limit of low-surface loading, we can only speculate about the behavior at high loading, in particular as far as the observed dominance of M is concerned. First, the intermolecular interactions between phosphates will take place through the phosphate OH groups which would decrease possibility of forming more covalent bonds between the surface and phosphates and Second, at higher phosphate loading, the Fe active sites will be less (with respect to the number of existing phosphate molecules) and then the chance for forming B motifs will decrease with respect to the M motifs. Since we have no computational support for this scenario, we did not add any comment to the manuscript.
---------------
Concerning the goethite-IHP interaction (paragraph 3.3.), the B motif is unstable and only three M motifs are observed: M(1), M(2) and 3M. The author suggested that the IHP adsorption decrease the interaction between water molecules and goethite surface probably because of the large size of the IHP molecule. I find that this phenomenon lacks an explanation. How the size of the IHP molecule can impact the surrounding water? IHP exhibits 22 hydrogen bonds with water (three time more than for GP molecule), thus IHP appears to have a strong hydration (e.g. outer-sphere complexation) shell which could affect surrounding water and hinders IHP adsorption.
ANSWER
We agree with the reviewer that in principle the competition with IHP hydration leading to outer sphere complexes deserves further studies. However, this would be a project on its own. To not confuse readers we have removed the sentence “In addition to this behavior, IHP has a bigger size compared to other phosphates and thus covers a larger part of the surface upon adsorption. Consequently, IHP hinders the attraction between water and the surface.
------
Minor comments:
Do the authors consider studying the effect of pH which seems to be an important parameter?
ANSWER
Thanks so much for this comment. We agree with the reviewer that pH plays a very important here. Therefore, we have already discussed this effect in a very recent study [1] related to orthophosphate adsorption at the goethite surface. Further, we are planning to study the pH effect for GP and IHP as well.
[1] Ahmed, Ashour A., Stella Gypser, Dirk Freese, Peter Leinweber, and Oliver Kuehn. “Molecular Level Picture of the Interplay between pH and Phosphate Binding at the Goethite–Water Interface.” Physical Chemistry Chemical Physics, 2020. https://doi.org/10.1039/D0CP04698A.
----------------
Did the authors observed a reorganization of the surface layers of goethite crystal?
ANSWER
We have not observed reorganization of the goethite surface layers.
------------------
I suggest to the authors to put the description of the IR spectrum in a table for easier reading.
ANSWER
We have already tabulated the IR spectra in Table 1 and provided the description using superscript symbols. We think that tabulating separately the calculated IR spectra would repeat the content in Table 1.
-----------------------------
Language comments:
p1 line 22 : “effect” → “affect” or “have an effect”
Done as suggested by the reviewer
p2 line 35 “these colloidals” → “these colloids”
Done as suggested by the reviewer
p2 line 37 “Orthophosphate (OP) is one of most...” → “Orthophosphate (OP) is one of the most...”
Done as suggested by the reviewer
p2 line 52: “outersphere complexes.” → “outer-sphere complexes.”
Done as suggested by the reviewer
p2 line 79 : “Some of the common goethite surface planes...” → “Some common goethite surface planes...”
Done as suggested by the reviewer
p3 line 86 : “pH less than the goethite PZC” → “pH lower than the goethite PZC”
Done as suggested by the reviewer
p3 line 94 : “covalent bond/s” → “covalent bonds”
We have changed “covalent bond/s” into “covalent bond(s)” and not into “covalent bonds”, since we wanted to say that GP has two options. It might form one bond (singular) or two bonds (plural) to have M or B motif, respectively.
p4 line 116 : “they should allow to draw conclusions” → “they should allow us to draw conclusions”
Done as suggested by the reviewer
p4 line 143 : “that differ based on level of theory” → “that differ based on the level of theory”
Done as suggested by the reviewer
p5 line 164 : “( ≈ 2TB)” → “( ≈ 2 TB)”
Done as suggested by the reviewer
p6 line 172, 193 and 195: “equlibration” → “equilibration”
Done as suggested by the reviewer
p6 line 190: “and an Fe–P distance” → “and Fe–P distance
Done as suggested by the reviewer
p6 line 195 : ”trajecotry” → “trajectory”
Done as suggested by the reviewer
p6 line 210: “Also, OP is know to form” → “Also, OP is known to form”
Done as suggested by the reviewer
p6 line 211: “the question why” → “the question of why”
Done as suggested by the reviewer
p7 line 216: “Neverthless” → “Nevertheless”
Done as suggested by the reviewer
p7 line 231: “within range” → “within the range”
Done as suggested by the reviewer
p8 line 247: “Also a proton transfer” → “Also, a proton transfer”
Done as suggested by the reviewer
p9 line 269: “to surface” → “to the surface”
Done as suggested by the reviewer
p9 line 273: . “Also one proton transfer” → “Also, one proton transfer”
Done as suggested by the reviewer
p9 line 290: “in latter case” → “in the latter case”
Done as suggested by the reviewer
p9 line 292: “covers larger part” → “covers a larger part”
Done as suggested by the reviewer
p9 line 293 : “Consquently” → “Consequently“
Done as suggested by the reviewer
p9 line 301: “one could also infer that that the changes...” → “one could also infer that that the changes...”
Done as suggested by the reviewer
p10 line 320: “(in parenthesis)” → “(in parentheses)”
Done as suggested by the reviewer
p11 line 366: “Regarding next band” → “Regarding next the band”
We have changed “Regarding next band” into “Regarding the next band”
p11 line 358, 373, 376, 377 and many other time in the manuscript: “strectching” → “stretching”
Done as suggested by the reviewer
p13 line 383: “betweens” → “between“
Done as suggested by the reviewer
p13 line 368: “comparision” → “comparison”
Done as suggested by the reviewer
p14 line 450, 451, 453: “assigned band” → “assigned the band”
Done as suggested by the reviewer
p14 line 455: “for same reasons” → “for the same reasons”
Done as suggested by the reviewer
p14 line 468: “The bands in frequency range” → “The bands in the frequency range”
Done as suggested by the reviewer
p14 line 471: Space missing at the beginning of the sentence
Done as suggested by the reviewer
p15 line 480: “Also its B motif” → “Also, its B motif”
Done as suggested by the reviewer
p15 line 490: “at surface” → “at the surface”
Done as suggested by the reviewer
p15 line 492: “at goethite surface” → “at the goethite surface”
Done as suggested by the reviewer
p15 line 494: Space missing at the beginning of the sentence
Done as suggested by the reviewer